# RNF213 promotes Treg cell differentiation by facilitating K63-linked ubiquitination and nuclear translocation of FOXO1

Xiaofan Yang[1,5], Xiaotong Zhu[2,5], Junli Sheng[1,5], Yuling Fu[1], Dingnai Nie[1], Xiaolong You[1], Yitian Chen[1], Xiaodan Yang[1], Qiao Ling[1], Huili Zhang[3] ✉, Xiaomin Li[4] ✉ & Shengfeng Hu [1,2] ✉

Autoreactive CD4[+] T helper cells are critical players that orchestrate the immune response both in multiple sclerosis (MS) and in other neuroinflammatory autoimmune diseases. Ubiquitination is a posttranslational protein modification involved in regulating a variety of cellular processes, including CD4[+] T cell differentiation and function. However, only a limited number of E3 ubiquitin ligases have been characterized in terms of their biological functions, particularly in CD4[+] T cell differentiation and function. In this study, we found that the RING finger protein 213 (RNF213) specifically promoted regulatory T (Treg) cell differentiation in CD4[+] T cells and attenuated autoimmune disease development in an FOXO1-dependent manner. Mechanistically, RNF213 interacts with Forkhead Box Protein O1 (FOXO1) and promotes nuclear translocation of FOXO1 by K63-linked ubiquitination. Notably, RNF213 expression in CD4[+] T cells was induced by IFN-β and exerts a crucial role in the therapeutic efficacy of IFN-β for MS. Together, our study findings collectively emphasize the pivotal role of RNF213 in modulating adaptive immune responses. RNF213 holds potential as a promising therapeutic target for addressing disorders associated with Treg cells.

Multiple sclerosis (MS) is a multifaceted autoimmune disorder affecting the central nervous system (CNS), characterized by degenerative processes typically manifesting between the ages of 20 and 40[1]. IFN-β is the primary therapeutic option for patients with a relapsing-remitting MS (RRMS), but its efficacy may vary among individuals[2]. Autoreactive CD4[+] T helper cells are critical players that orchestrate the immune response both in MS and in other neuroinflammatory autoimmune diseases including experimental autoimmune encephalomyelitis (EAE), a kind of animal model that has been developed for MS[3]. The CD4[+] T cells are the primary cellular components involved in the acquired immune response

and play a crucial role in body development and homeostasis. The differentiation of naïve CD4[+] T cells into various helper T (Th) cell subsets, including Th1, Th2, Th17, regulatory T (Treg) cells, and other subsets, initiates a cascade of immune response processes[4]. The involvement of Th1 and Th17 cells in the pathogenesis of autoimmune diseases has been well-documented[5,6]. In particular, Th17 cells have emerged as key mediators driving autoimmunity and subsequent tissue damage[7]. Treg cells represent a distinct subset of CD4[+] T cells that play an indispensable role in maintaining immune system homeostasis, restraining self-reactivity, and modulating exaggerated immune

[1]The Second Affiliated Hospital, The Second School of Clinical Medicine, The State Key Laboratory of Respiratory Disease, Guangdong Provincial Key Laboratory of Allergy & Clinical Immunology, Guangzhou Medical University, Guangzhou, China. [2]Department of Rheumatology and Clinical Immunology, Zhujiang Hospital, Southern Medical University, Guangzhou, China. [3]Department of Neurology, School of Medicine, Guangzhou First People's Hospital, South China University of Technology, Guangzhou, China. [4]Department of Respiratory, Guangzhou Women and Children's Medical Center, Guangzhou Medical University, Guangzhou, China. [5]These authors contributed equally: Xiaofan Yang, Xiaotong Zhu, Junli Sheng. ✉e-mail: zhanghuili211@163.com; Xiaomin_lilee@163.com; hushengfeng@gzhmu.edu.cn

responses towards foreign antigens[8,9]. The expression of the transcription factor forkhead box P3 (Foxp3) serves as a defining feature of Treg cells and functions as the pivotal regulator governing Treg cell differentiation, maintenance, and functionality[10]. In vivo, Treg cells can develop in the thymus during positive selection, and these cells have been termed thymus-derived Tregs (tTreg), but also they may be generated from naïve CD4+ T cells after stimulation in the presence of TGF-β and IL-2 in the periphery (periphery Treg, pTreg)[9,11]. Naive CD4+ T cells can be induced to differentiate into Treg cells in vitro through stimulation with TGF-β and IL-2, thus these cells are commonly referred to as induced Treg cells (iTreg)[10]. The immunosuppressive function of Treg cells is primarily mediated by the cytokines TGF-β and IL-10[12]. Forkhead Box Protein O1/3a (FOXO1/3a) proteins play a vital role in the early differentiation of Treg cell lineage, and nuclear localization of FOXO1/3a facilitates the expression of Foxp3 and enhances Treg cell stability[13]. The subsets of CD4+ T cells exhibit distinct developmental and regulatory mechanisms, the understanding of which would greatly contribute to the advancement of therapies for inflammatory diseases or the resolution of infections caused by pathogens.

Ubiquitination is a posttranslational protein modification involved in regulating a variety of cellular processes, including CD4+ T cell differentiation and function[14]. Ubiquitin (Ub) chains are assembled in a three-step enzymatic reaction carried out by Ub-activating enzymes (E1), Ub-conjugating enzymes (E2), and Ub-protein ligases (E3)[15]. E3 ligase determines substrates by associating with target proteins[16]. The formation of various types of polyubiquitin chains occurs by linking the C-terminal glycine of ubiquitin to any one of the seven internal lysine residues in the preceding ubiquitin[17]. Different types of poly-Ub chains serve distinct functions: K48-linked poly-Ub chains specifically target substrates for proteasomal degradation, while K63-linked poly-Ub chains primarily mediate non-degradative signaling processes[18]. The RING finger (RNF) protein, encompassing the RING domain, represents the largest family of E3 ubiquitin ligases with 340 validated members in humans[19]. Various RNF ubiquitin ligases, play pivotal roles in coordinating the immune response by ensuring the proper functioning of diverse cell populations[20–24]. Our previous study found that RNF157 attenuates CD4+ T cell-mediated autoimmune response by promoting HDAC1 ubiquitination and degradation[25]. However, only a limited number of E3 ubiquitin ligases have been characterized in terms of their biological functions, particularly in CD4+ T cell differentiation and function.

RNF213 is a very large, poorly characterized E3 ubiquitin ligase. The RNF213 gene has been identified as a susceptibility gene for moyamoya disease, a cerebrovascular disorder that predominantly affects populations of East Asian descent[26,27]. Recent functional studies highlighted the role of RNF213 in lipid metabolism, mediating lipid droplet formation, and lipotoxicity[28,29]. RNF213 also plays an important role in host defense against various pathogenic microorganisms including herpes virus[30], *Salmonella*[31], and *Listeria monocytogenes*[32]. Notably, RNF213 expression was induced by IFN-β and mediated antiangiogenic activity or antimicrobial activity of IFN-β[32,33]. However, RNF213 also referred to as mysterin[33], possesses numerous uncharacterized functionalities.

In this study, we find that RNF213 specifically promotes Treg cell differentiation in CD4+ T cells and attenuates EAE development in a FOXO1-dependent manner. Mechanistically, RNF213 promotes Foxp3 expression by interacting with FOXO1 and promoting K63-linked ubiquitination and nuclear translocation of FOXO1. Notably, RNF213 expression in CD4+ T cells is induced by IFN-β and exerts a crucial role in the therapeutic efficacy of IFN-β for EAE and MS.

## Results
### RNF213-regulated differentiation of CD4+ T cell subpopulations during autoimmunity
To understand the biological function of RNF213 in the CD4+ T cell responses, we first generated mice with RNF213-deficient (*Rnf213−/−*).

We found that *Rnf213−/−* mice did not show any distinct abnormalities in thymic or peripheral T cell homeostasis (Supplementary Fig. 1A–G), including intracellular IFN-γ and IL-17A expression, and tTreg cell differentiation in splenic CD4+ T cells (Supplementary Fig. 1H, I). Subsequently, we induced EAE in wild-type (WT) and *Rnf213−/−* mice after immunization with the myelin oligodendrocyte glycoprotein (MOG) (35-55) peptide. The results showed that *Rnf213−/−* developed more severe EAE, with higher clinical score (Supplementary Fig. 2A) and increased proinflammatory cytokines, IFN-γ, IL-17A, and GM-CSF, but decreased IL-10 in serum (Supplementary Fig. 2B) when compared to WT control mice. Analysis of the CD4+ T cell population in the central nervous system (CNS) revealed that *Rnf213−/−* mice had increased numbers of IFN-γ+ and IL-17A+ CD4+ T cells (Supplementary Fig. 2C) and decreased numbers of Treg cells (CD25+ Foxp3+) (Supplementary Fig. 2D). Consistently, after stimulation with MOG(35-55) peptide, infiltrating cells from *Rnf213−/−* mice secreted higher levels of IFN-γ and IL-17A and lower levels of IL-10 (Supplementary Fig. 2E).

To gain a deeper comprehension of the influence exerted by RNF213 on the regulation of CD4+ T cell immune responses, *Rag1−/−* recipient mice were administered WT and *Rnf213−/−* naïve CD4+ T cells, respectively, and were subsequently induced to EAE. *Rag1−/−* mice receiving *Rnf213−/−* CD4+ T cells developed significantly severe EAE compared to those receiving WT CD4+ T cells (Supplementary Fig. 2F). The levels of IFN-γ and IL-7A were higher, but the level of IL-10 was lower in the serum of *Rag1−/−* recipients receiving *Rnf213−/−* CD4+ T cells than those in the serum of those receiving WT CD4+ T cells (Supplementary Fig. 2G). *Rag1−/−* mice receiving *Rnf213−/−* CD4+ T cells had a higher proportion of Th1 and Th17 in the CNS than those that received WT CD4+ T cells (Supplementary Fig. 2H), but a lower proportion of Treg cells (Supplementary Fig. 2I).

To further analyze the function of RNF213 in CD4+ T cells, we generated mice with conditional RNF213 knockout in CD4+ T cells (*Rnf213fl/fl CD4Cre*) through crossing *Rnf213flox/flox* (*Rnf213fl/fl*) and CD4-Cre mice. *Rnf213fl/fl CD4Cre* mice did not show any distinct abnormalities in thymic or peripheral T cell homeostasis (Supplementary Fig. 3A–G), including intracellular IFN-γ and IL-17A expression, and Treg cell differentiation in splenic CD4+ T cells (Supplementary Fig. 3H, I). Consistence with *Rnf213−/−* mice, conditional RNF213 knockout in CD4+ T cells promoted EAE progression (Fig. 1A). Compared with *Rnf213fl/fl* mice, *Rnf213fl/fl CD4Cre* mice produced higher serum levels of IFN-γ, IL-17A and GM-CSF, but lower serum levels of IL-10 (Fig. 1B). After MOG(35-55) peptide immunization, CD4+ T cells in the CNS of *Rnf213fl/fl CD4Cre* mice showed increased intracellular expression of IFN-γ and IL-17A compared with those of *Rnf213fl/fl* mice (Fig. 1C). After RNF213 deficiency, the proportion of Treg within the CD4+ T cell population exhibited a significant decrease (Fig. 1D). Overall, these results indicated that RNF213 might serve a critical role in autoimmunity through regulating the differentiation of CD4+ T cell subpopulations.

### RNF213 promoted Treg cell differentiation in CD4+ T cells
We further determined the role of RNF213 in regulating CD4+ T cells in vitro. We first examined the endogenous expression of RNF213 in CD4+ T cell subpopulations. The results revealed a significant downregulation of RNF213 expression at both mRNA and protein levels across all differentiated subpopulations, except for Treg cells where the reduction was minimal (Supplementary Fig. 4A, B). The expression of the activation markers CD69 and cell proliferation was not affected by the deletion of RNF213 following treatment with anti-CD3 and anti-CD28 antibodies (Supplementary Fig. 4C, D). To assess the role of RNF213 in the regulation of CD4+ T cell differentiation, we differentiated WT and *Rnf213−/−* naïve CD4+ T cells under different polarizing conditions. We observed comparable frequencies of IFN-γ+ CD4+ T cells and levels of IFN-γ between WT and *Rnf213−/−* CD4+ T cells under Th1-cell differentiation conditions (Supplementary Fig. 4E),

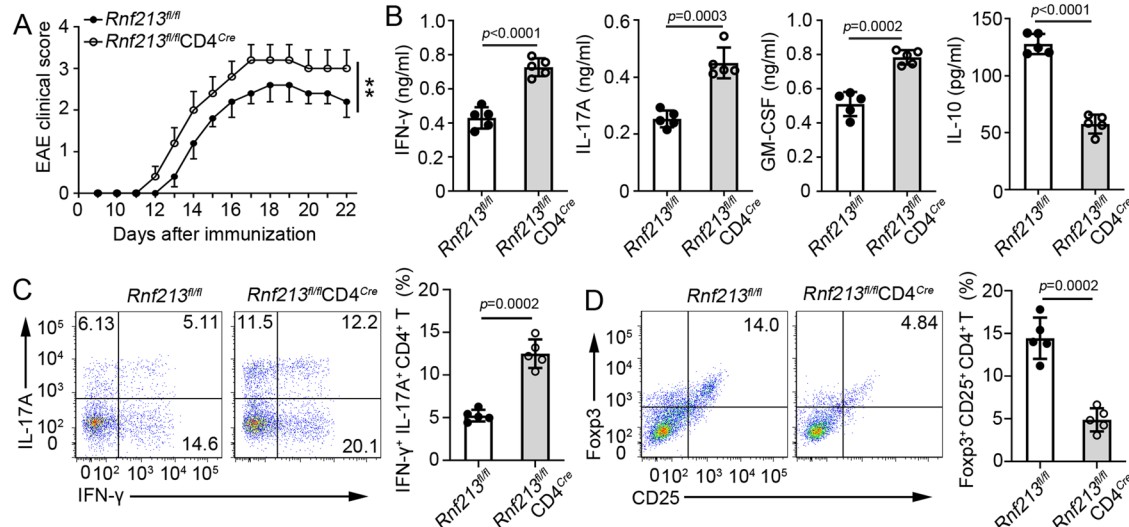

**Fig. 1 | RNF213 deficiency in CD4+ T cells promoted EAE development and inhibiting Treg cell differentiation.** *Rnf213fl/fl* and *Rnf213fl/flCD4Cre* mice were immunized with MOG(35-55) peptide in CFA adjuvant and pertussis toxin to induce EAE. **A** The graph shows the clinical score of EAE (n = 5 per group). **B** Concentration of IFN-γ, IL-17, GM-CSF, and IL-10 in serum was measured by ELISA on day 20. **C** The cells infiltrating the central nervous system (CNS) were re-stimulated with MOG(35-55) peptide directly ex vivo, and the intracellular production of IFN-γ and IL-17A by CD4+ T cells was determined. Pooled data are presented in the right panel. **(D)** Flow cytometry of Treg cell (CD25+ Foxp3+) in CNS of *Rnf213fl/fl* and *Rnf213fl/flCD4Cre* mice on days 20 during EAE. Pooled data are presented in the right panel. Data shown are the mean ± SD. *P < 0.05, **P < 0.01, and ***P < 0.001. P values were calculated using a two-sided, unpaired Student's t-test (**A**–**D**). N = 5 (**B**–**D**) repeats from three independent experiments. Source data are provided as a Source Data file.

as well as similar frequencies of IL-17A+ CD4+ T cells and levels of IL-17A under Th17-cell differentiation conditions (Supplementary Fig. 4F). However, RNF213 deficiency resulted in a significantly lower frequency of Foxp3+ CD4+ T cells and increased IL-10 production after Treg skewing (Supplementary Fig. 4G). Compared to WT CD4+ T cells, *Rnf213−/−* CD4+ T cells also expressed lower levels of *Foxp3* mRNA in the presence of TGF-β (Supplementary Fig. 4H). Reintroduction of RNF213 into WT and *Rnf213−/−* CD4+ T cells restored Treg differentiation (Supplementary Fig. 4I).

To explore the intrinsic role of RNF213 in the differentiation of Treg cells in vivo, competitive adoptive CD4+ T cell transfer assays were performed. *Rag1−/−* recipient mice received CD45.1+ WT and CD45.2+ *Rnf213−/−* naïve CD4+ T cells (1:1) and were subsequently induced to EAE (Fig. 2A). CD4+ T cell percentages of CD45.1+ WT and CD45.2+ *Rnf213−/−* cells showed no significant difference (Fig. 2B). Likewise, in the same environment RNF213 deficiency in CD4+ T cells no longer affected Th1 and Th17 differentiation (Fig. 2C). However, compared to CD45.1+ WT CD4+ T cells, less CD45.2+ *Rnf213−/−* CD4+ T cells differentiated into Foxp3+CD25+ cells (Fig. 2D). CD45.2+ *Rnf213−/−* CD4+ T cells expressed less mRNAs of *Foxp3*, *Tgfb1* and *Il10* than CD45.1+ WT CD4+ T cells did (Fig. 2E). Taken together, these results suggested that RNF213 plays a specific role in promoting the differentiation of Tregs within CD4+ T cell population.

## RNF213 deficiency in CD4+ T cells promoted autoimmunity by inhibiting immunosuppressive activity of Treg cell

To investigate the impact of RNF213-deficient Treg cells on the immune response, *Rag1−/−* recipient mice were intravenously injected with WT CD45.1+ naïve CD4+ T cells along with sorted Treg cells from either WT or *Rnf213−/−* mice, followed by induction of EAE (Fig. 3A). The results showed that recipient mice that received WT CD45.1+ naïve CD4+ T and *Rnf213−/−* Treg cells developed modestly severe EAE (Fig. 3B) with an increased IL-17A and a decreased IL-10 in the serum compared to recipient mice that received WT CD45.1+ naïve CD4+ T and WT Treg (Fig. 3C). The proportion and quantity of CD45.1+ CD4+ T cells infiltrating the CNS in recipient mice that received WT CD45.1+ naïve CD4+

T cells and *Rnf213−/−* Treg cells were significantly increased compared to those in recipient mice that received WT CD45.1+ naïve CD4+ T cells and WT Treg cells (Fig. 3D), as well as intracellular expression of IFN-γ and IL-17A (Fig. 3E). The proportion of Treg cells within the CD45.1+ CD4+ T cell population from recipient mice that received WT CD45.1+ naïve CD4+ T and WT Treg cells exhibited a significant increase, although the total number was reduced (Fig. 3F). 2D2 mice belong to MOG(35-55)-specific T cell receptor transgenic mice, and adoptive transfer of its CD4+ T cells to *Rag1−/−* recipients can spontaneously induce the generation of EAE phenotypes. We adopted naïve CD4+ T cells from 2D2 mice with sorted Treg cells from either WT or *Rnf213−/−* mice into *Rag1−/−* recipients (Supplementary Fig. 5A). Likewise, recipient mice that received 2D2 CD4+ T and *Rnf213−/−* Treg cells developed modestly severe EAE (Supplementary Fig. 5B) with an increased IL-17A and a decreased IL-10 in the serum compared to recipient mice that received 2D2 CD4+ T and WT Treg (Supplementary Fig. 5C).

To further explore the effect of RNF213 on the immunosuppressive activity of Treg cells, CD45.2+ WT or *Rnf213−/−* Treg cells were enriched and co-cultured with different ratios of CD45.1+ WT CD4+ T cells that had been activated by anti-CD3 and anti-CD28 antibodies. The proliferation of CD45.1+ CD4+ T lymphocytes or CD45.2 Treg was assessed using a CFSE dilution assay. The results showed that Treg cells from *Rnf213−/−* mice had a reduced capacity to inhibit the proliferation of CD4+ T cells (Supplementary Fig. 5D). However, RNF213 deficiency in Treg cells would impede their proliferation levels (Supplementary Fig. 5D). Together, these results suggested that RNF213 deficiency in CD4+ T cells promoted autoimmunity by inhibiting immunosuppressive activity of Treg cells.

## RNF213 interacts with FOXO1 and facilitates the nuclear translocation of FOXO1

To investigate the molecular mechanism by which RNF213 regulates Treg differentiation, we performed a mass spectrometric analysis to identify RNF213-interacting proteins in Treg cells from WT or RNF213-Tg mice (*Rnf213TgFoxp3Cre*), in which Flag-tagged RNF213 was conditionally overexpressed in Treg cells (Fig. 4A). One of the strongest RNF213-interacting proteins identified was

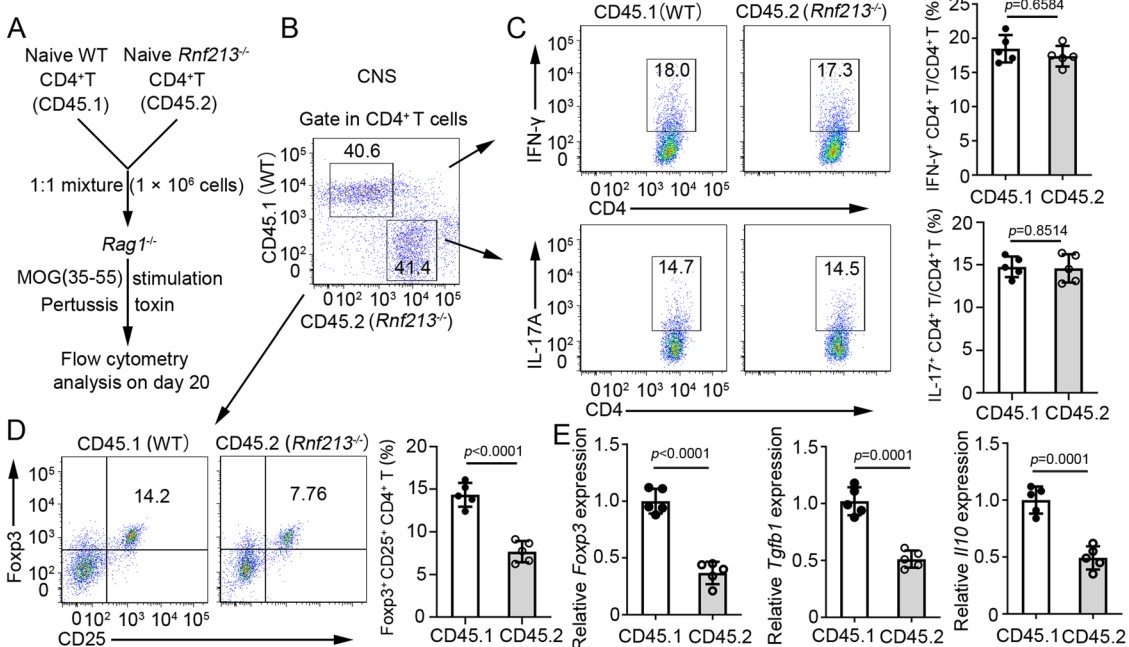

**Fig. 2 | RNF213 promoted Treg cell differentiation in a T cell-intrinsic manner in vivo. A** Schematic of experimental design of competitive adoptive CD4⁺ T cell transfer assays. **B** The cells infiltrating the central nervous system (CNS) were harvested, and the percentages of WT (CD45.1⁺) and *Rnf213⁻/⁻* (CD45.2⁺) in the CD4⁺ T cell populations were determined. **C, D** Flow cytometry of IFNγ⁺, IL-17A⁺ CD4⁺ cells (**C**) or CD4⁺ CD25⁺ Foxp3⁺ Treg cells (**D**) in CNS of WT and *Rnf213⁻/⁻* mice on days 20

during EAE. **E** QPCR analysis of Foxp3, TGF-β1, and IL-10 in CD4⁺ cells isolated from CNS of WT and *Rnf213⁻/⁻* mice. Data shown are the mean ± SD. *$P < 0.05$, **$P < 0.01$, and ***$P < 0.001$. *P* values were calculated using a two-sided, unpaired Student's *t*-test in (**C–E**). N = 5 (**C–E**) repeats from three independent experiments. Source data are provided as a Source Data file.

FOXO1 (Fig. 4B, C), which was validated by endogenous co-immunoprecipitation (CO-IP) in Treg cell (Fig. 4D) or exogenous CO-IP in HEK293 T cells (Fig. 4E). The immunofluorescence results also demonstrated the co-localization of RNF213 and FOXO1 (Supplementary Fig. 6A). The translocation of FOXO1 into the nucleus and its subsequent binding to the transcriptional region of Foxp3 is indispensable for initiating Treg cell differentiation[13]. Luciferase results showed that RNF213 enhanced the activity of the Foxp3 promoter in a dose-dependent manner (Fig. 4F). FOXO1 mRNA and protein expression were not affected by RNF213 deficiency under Treg cell differentiation conditions within a short time (Supplementary Fig. 6B, C). Therefore, we hypothesize whether RNF213 may exert regulatory control over the nuclear translocation of FOXO1. The immunofluorescence results showed a significant reduction in the nuclear translocation of FOXO1 due to RNF213 deficiency (Fig. 4G). Consistent with these results, the nuclear translocation of FOXO1 was diminished in *Rnf213⁻/⁻* Treg cells as evidenced by Western blot analysis (Supplementary Fig. 6D). These results demonstrated that RNF213 interacts with FOXO1 and facilitates the nuclear translocation of FOXO1.

### RNF213-regulated Treg cell differentiation depends on FOXO1

To assess whether FOXO1 is the primary target of RNF213 in regulating Treg cell differentiation, mice conditional overexpression for RNF213 but knockout for FOXO1 in Treg cells (*Rnf213^Tg^Foxo1^fl/fl^Foxp3^Cre^*) were generated. *Rnf213^Tg^*, *Rnf213^Tg^Foxp3^Cre^*, *Foxo1^fl/fl^*, *Foxo1^fl/fl^Foxp3^Cre^*, and *Rnf213^Tg^Foxo1^fl/fl^Foxp3^Cre^* mice were induced to EAE (Fig. 5A). The EAE score and serum levels of IL-17 and IL-10 were significantly different between *Rnf213^Tg^* and *Rnf213^Tg^Foxp3^Cre^* mice. However, this difference was not observed when mice were also deficient in FOXO1 (Fig. 5B, C). However, this difference was not observed when mice were deficient in FOXO1 (Fig. 5B, C). Similarly, no difference in Treg cell differentiation was observed between the groups deficient in FOXO1, regardless of whether RNF213 was also overexpressed (Fig. 5D). These results

demonstrated that RNF213 promoted Treg differentiation in a FOXO1-dependent manner.

### RNF213-mediated K63-linked ubiquitination of FOXO1

Given the dependence of RNF213-regulated Treg cell differentiation on FOXO1, we sought to investigate the potential role of RNF213, an E3 ubiquitin ligase, in mediating FOXO1 function through poly-ubiquitination. To examine the type of RNF213-mediated ubiquitin linkage of FOXO1, WT ubiquitin, and mutants with a single lysine changed to arginine (K6R, K11R, K27R, K29R, K33R, K48R, and K63R) were used in in vitro ubiquitination assays. The results showed that Lys63 was necessary for RNF114-mediated polyubiquitin chain assembly (Fig. 6A). The ubiquitination assays in Treg cells showed that K63-linked ubiquitination of FOXO1 was significantly decreased in RNF213-deficient Treg cells (Supplementary Fig. 7A). To map the interaction regions of FOXO1 and RNF213, we expressed a series of truncated forms of FOXO1 in HEK293T cells, followed by CO-IP assays. The results showed that the forkhead domain (FHD) of FOXO1 was essential for its interaction with RNF213 (Supplementary Fig. 7B). Furthermore, K207, K242 and K245 were predicted as the E3-specific ubiquitination sites of FOXO1 according to the UbiqSite Database (http://systbio.cau.edu.cn/ubiqsite/index.php) or previous report[34]. However, site-mapping analysis revealed that mutation of K207 into arginine residues (K207R) impaired the RNF213-mediated ubiquitination of FOXO1 (Supplementary Fig. 7C). Luciferase results showed that K207R mutation of FOXO1 impaired RNF213-mediated the activity of the Foxp3 promoter (Supplementary Fig. 7D). The overexpression of FOXO1 in *Rnf213⁻/⁻* naive CD4⁺ T cells exhibited minimal rescue effects on the generation of Treg cells, whereas the overexpression of FOXO1 in *Rnf213^Tg^CD4^Cre^* (in which Flag-tagged RNF213 was conditional overexpressed in CD4⁺ T cells) CD4⁺ T cells significantly enhanced the generation of Treg cells (Supplementary Fig. 7E). However, the overexpression of FOXO1 with K207R mutation had limited impact on improving the generation of Treg cells (Supplementary Fig. 7E).

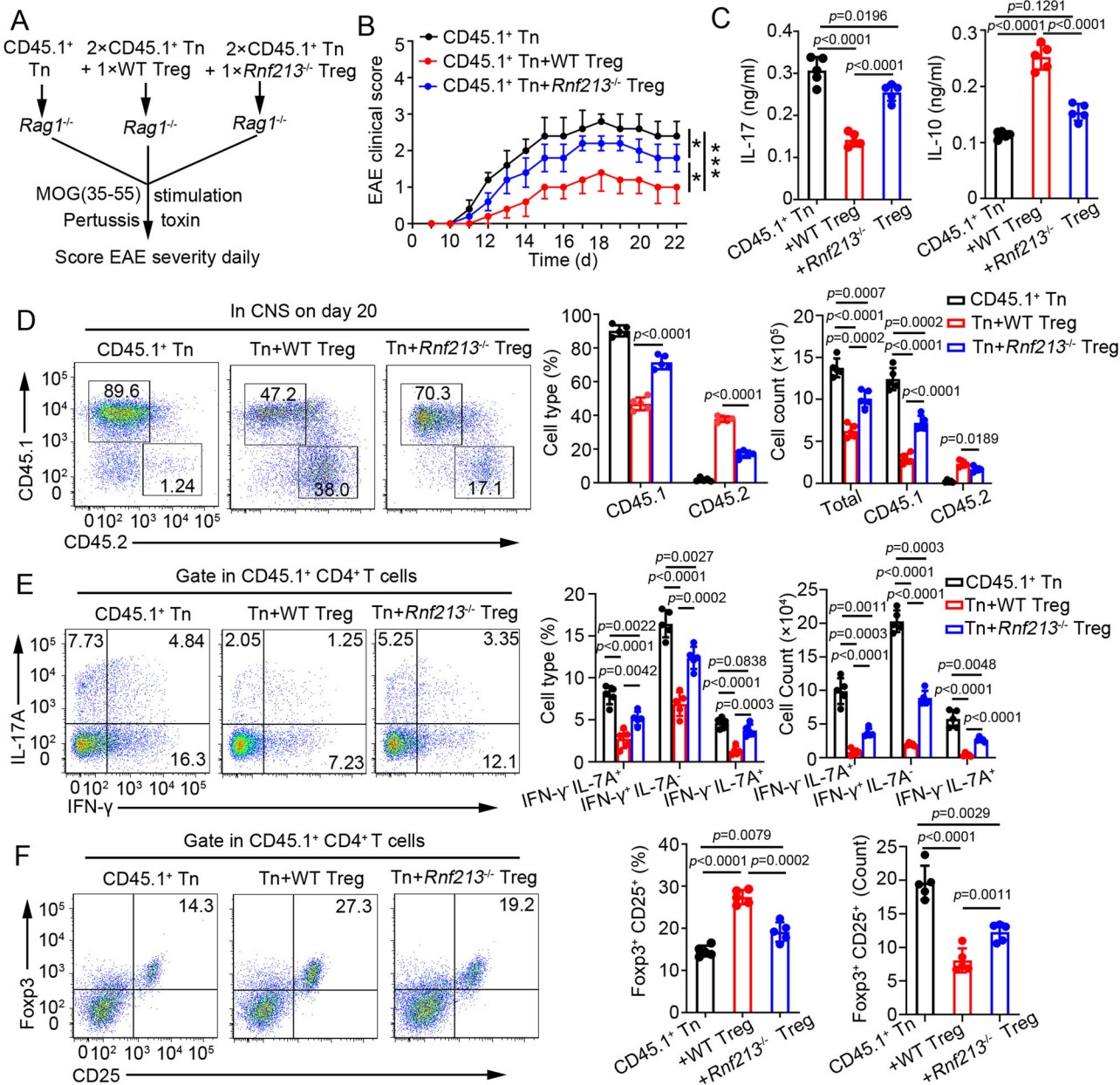

**Fig. 3 | RNF213 is essential for Treg cells to suppress T cell-mediated EAE.**
**A** Schematic of experimental design of adoptive CD4⁺ T cell transfer assays. In detail, $Rag1^{-/-}$ mice were transferred with $1 \times 10^6$ CD44⁻ CD62L⁺ CD4⁺ T cells (Tn) from CD45.1⁺ mice alone (n = 5) or in combination with Tn from CD45.2⁺ WT or $Rnf213^{-/-}$ Tregs. **B** The graph shows the clinical score of EAE (n = 5 per group). **C** Concentration of IL-17 and IL-10 in serum was measured by ELISA on day 20. **D** The cells infiltrating the central nervous system (CNS) were harvested, and the percentages of CD45.1⁺ and CD45.2⁺ in the CD4⁺ T cell populations was determined.

Pooled data of the percentages and cell counts are presented in the right panel. **E, F** Flow cytometry of IFNγ⁺, IL-17A⁺ CD4⁺ cells (**E**) or CD4⁺ CD25⁺ Foxp3⁺ Treg cells (**F**) in CNS of recipient mice on days 20 during EAE. Pooled data of the percentages and cell counts are presented in the right panel. Data shown are the mean ± SD. Ns, no significance, *$P < 0.05$, **$P < 0.01$, and ***$P < 0.001$. $P$ values were calculated using a two-sided, unpaired Student's $t$-test in (**B**–**F**). N = 5 (**C**–**F**) repeats from three independent experiments. Source data are provided as a Source Data file.

Furthermore, we observed that the RING domain of RNF213 (which mediates the ubiquitination activity of RNF213) played a crucial role in its interaction with FOXO1 (Fig. 6B). WT RNF213 increased FOXO1-mediated activity of the Foxp3 promoter, but a truncated form of FOXO1 lacking the RING domain did not (Fig. 6C). Consistent with that, overexpression of WT RNF213 in naive CD4⁺ T cells resulted in a significantly higher frequency of Foxp3⁺ cells under Treg-skewing conditions, but overexpression of the truncated form of RNF213 lacking the RING domain did not (Fig. 6D). These results suggested that RNF213 promoted Treg cell differentiation through regulating K63-linked ubiquitination of FOXO1.

## RNF213 promoted Treg cell differentiation in human CD4⁺ T cells

Next, we sought to clarify whether RNF213 regulates human CD4⁺ T cell functions through the same molecular mechanism. Similarly, all differentiated Th subpopulations had a significant downregulation of RNF213 expression at both mRNA and protein levels, with the exception of Treg cells where the reduction was minimal (Fig. 7A, B). The overexpression of RNF213 in CD4⁺ T cells did not affect the expression of the activation markers CD69, cell proliferation, and Th1 and Th17 differentiation (Supplementary Fig. 8A, B). Correspondingly, overexpression of RNF213 in human CD4⁺ T cells promoted FOXO1-

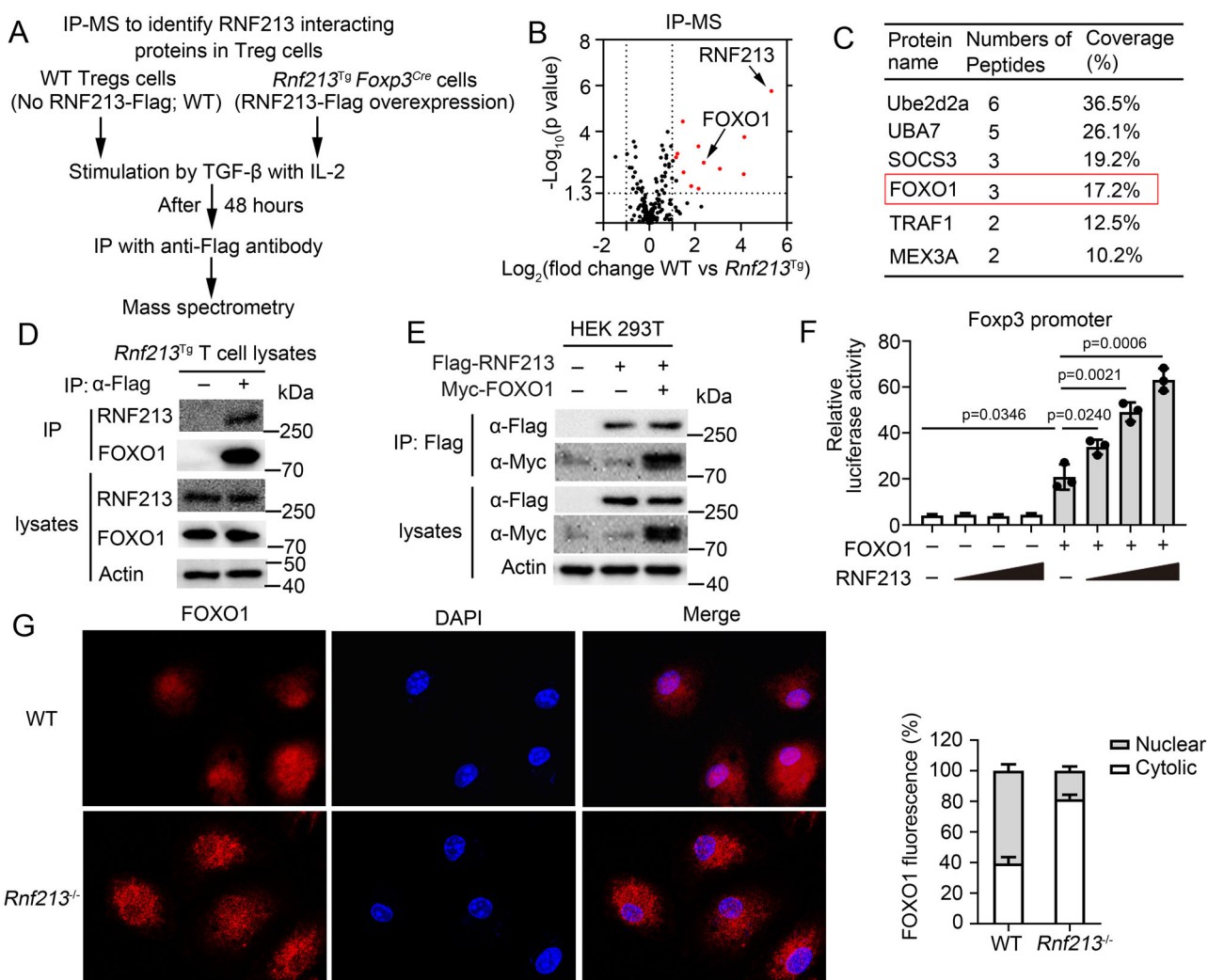

**Fig. 4 | RNF213 interacts with FOXO1 and is critical to regulate the translocation of FOXO1 to the nucleus. A** Schema of IP-MS approach to identify RNF213-interacting proteins in Treg cells. **B** Volcano plot of RNF213-interacting proteins by immunoprecipitation–mass spectrometry (IP-MS) analysis. Red indicates significantly enriched proteins ($\log_2$(fold change) >1; $t$-test adjusted $P < 0.05$). **C** The list of Ube2d2a, UBA7, SOCS3, FOXO1, TRAF1 and MEX3A identified by IP-MS analysis. **D** Immunoprecipitation (IP) and immunoblot (IB) analysis of Treg cells isolated from WT or $Rnf213^{Tg}$ mice. **E** IP and IB analysis of HEK293 cells transfected with the indicated plasmids for 24 h. **F** Luciferase activity of HEK293T cells transfected with a luciferase reporter driven by the Foxp3 promoter and expressing vector alone or various combinations (horizontal axis) of FOXO1 and RNF213. **G** Treg cells were double-stained with anti-FOXO1 (red) Abs and DAPI (nucleus, blue) and then observed by fluorescence microscopy. Scale bars: 50 μm. Quantification of fluorescent digital imaging analysis of FOXO1 expression in the cytosol and the nucleus depicted as the percentage expression levels are presented in the right panel ($n = 3$). Data shown are the mean ± SD. *$P < 0.05$, **$P < 0.01$, and ***$P < 0.001$. $P$ values were calculated using a two-sided, unpaired Student's $t$-test in (**F**). N = 3 (**E**) repeats from three independent experiments. Source data are provided as a Source Data file.

mediated activity of the Foxp3 promoter, a significantly increased frequency of Foxp3$^+$ cells under Treg-skewing conditions, and higher levels of *Foxp3* mRNA in the presence of TGF-β (Fig. 7C–E). However, no such effects were observed with overexpression of RNF213ΔRING (Fig. 7C–E). Given that RNF213 regulates Treg cells to inhibit the development of EAE, we sought to investigate its relevance in MS patients. RNA-seq (GSE66763) and qPCR results showed that CD4$^+$ T cells from healthy control individuals expressed high amounts of RNF213, whereas expression was reduced in CD4$^+$ T cells from patients with MS, especially in MOG-specific CD4$^+$ T cells (Fig. 7F, G). Taken together, these results suggested that RNF213 facilitates the differentiation of Treg cells in human CD4$^+$ T cells.

**The involvement of RNF213 is crucial in the therapeutic efficacy of IFN-β for EAE**

Considering the established efficacy of IFN-β as a therapeutic intervention for MS[2] and the role of RNF213 as an inducer of IFN-I[32,33], we

aimed to investigate the potential involvement of RNF213 in IFN-β therapy for MS. First, we examined RNA sequencing (RNA-seq) data (GSE195541) from the NCBI Gene Expression Omnibus (GEO)[35] to identify RNF213 expression pattern in CD4$^+$ T cells relevant to IFN-β stimulation. Following IFN-β stimulation, there was a significant increase in the expression of RNF213 within CD4$^+$ T cells (Supplementary Fig. 9A), which is consistent with QPCR results (Supplementary Fig. 9B). We then cultured naive CD4$^+$ T cells from WT or $Rnf213^{-/-}$ mice with under Treg cell polarizing conditions in the presence or absence of IFN-β and quantified the frequency of Foxp3$^+$ cells. These assays showed a dose-dependent, IFN-β-induced augmentation of the frequency of Foxp3$^+$ cells, while the effect was inhibited by RNF213 deficiency (Supplementary Fig. 9C). Furthermore, we investigate the contribution of RNF213 to the therapeutic efficacy of IFN-β for EAE (Supplementary Fig. 9D). The results showed that $Rnf213^{fl/fl}$ mice that received IFN-β treatment developed modestly attenuated EAE (Supplementary Fig. 9E) with a decreased IL-17A and an increased IL-10 in

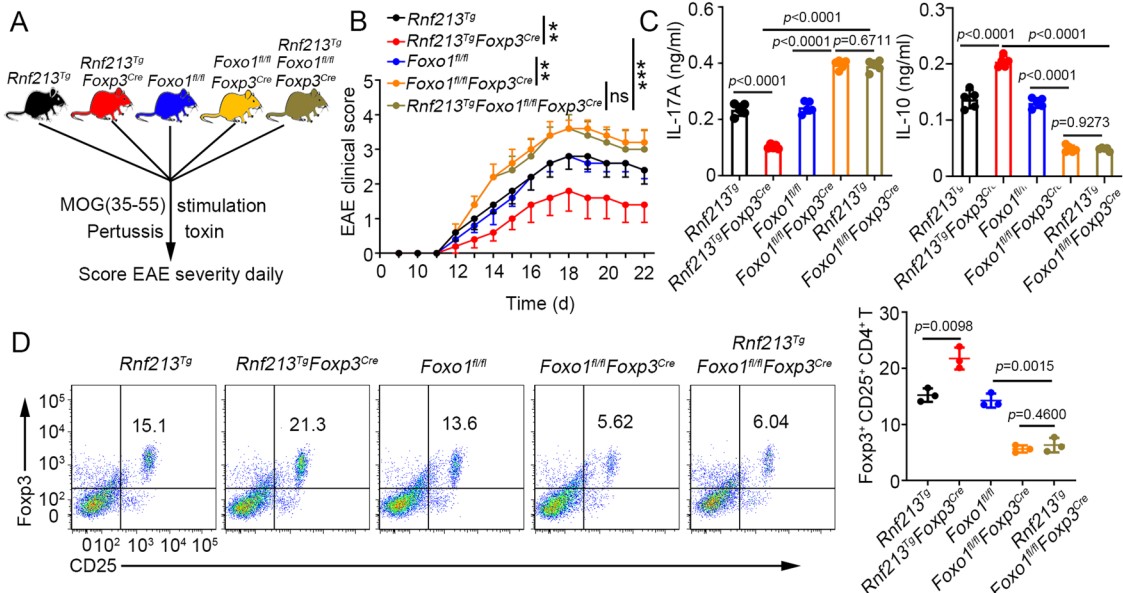

**Fig. 5 | RNF213-regulated Treg differentiation in a FOXO1-dependent manner.**
**A** Schematic of experimental design for multiple transgenic mice. **B** The graph shows the clinical score of EAE (n = 5 per group). **C** Concentration of IL-17 and IL-10 in serum was measured by ELISA on day 20. **D** Flow cytometry of Treg cell (CD25+ Foxp3+) in CNS on days 20 during EAE. Pooled data are presented in the right panel. Data shown are the mean ± SD. *P < 0.05, **P < 0.01, and ***P < 0.001. P values were calculated using a two-sided, unpaired Student's t-test in (**B**–**D**). N = 5 (**C**), n = 3 (**D**) repeats from three independent experiments. Source data are provided as a Source Data file.

the serum compared to control mice (Supplementary Fig. 9F). IFN-β can elicit an increase in Treg cells, however, the absence of RNF213 results in a near-complete abrogation of this effect (Supplementary Fig. 9G). Although IFN-β has been a first-line treatment for MS for over 20 years in RRMS, but it is not effective in 7-49% of patients with RRMS[36]. Treatment with IFN-β did enhance the differentiation of human CD4+ T cells into Treg cells (Supplementary Fig. 9H). The next step is to investigate the potential association between RNF213 and IFN-β in response to therapy for MS. QPCR results showed that RNF213 expression was higher in CD4+ T cells from MS patients who exhibited a positive response to IFN-β and further increased following IFN-β stimulation, whereas the expression of RNF213 was lower in non-responsive MS patients and remained unchanged after IFN-β stimulation (Supplementary Fig. 9I). Taken together, these results suggested that RNF213 facilitates the differentiation of Treg cells and exerts a crucial role in the therapeutic efficacy of IFN-β for MS.

## Discussion

Autoreactive CD4+ T helper cells play a crucial role in orchestrating the immune response, not only in MS but also in other neuroinflammatory autoimmune diseases[3]. Additionally, ubiquitination plays a vital role in the differentiation and function of CD4+ T cells[14]. Therefore, this study focuses on elucidating the developmental and regulatory mechanisms underlying CD4+ T cell responses during MS. We identified RNF213 as a regulator of Treg cell differentiation. In CD4+ T cells, RNF213 promoted K63-linked ubiquitination and nuclear translocation of FOXO1, thereby promoting the differentiation of Treg cells and mitigating the development of EAE (Supplementary Fig. 10).

Several studies have shown that RNF213 plays a vital role in a variety of physiological functions. To investigate the role of RNF213 in CD4+ T cell differentiation and function, we generated whole-body RNF213 knockout mice or conditionally knocked out RNF213 specifically in CD4+ T cells. However, no discernible disparity was observed in the regulatory function of RNF213 on the differentiation of CD4+ T cells, regardless of whether RNF213 is systemically knocked out or specifically knocked out in CD4+ T cells (Fig. 1 and Supplementary Fig. 2). These results suggested that the deficiency of

RNF213 in other cell types has minimal impact on the differentiation of CD4+ T cells. Despite there is a study demonstrating that dysregulation of RNF213 contributes to T cell response through antigen uptake, processing, and presentation[37]. A previous study showed that RNF213 was also involved in the cell proliferation of endothelial cells through decreasing AKT phosphorylation and inducing matrix metalloproteinase-1[38]. However, our results did not show that RNF213 inherently regulated T cell proliferation in CD4+ T cells; instead, it exerts an inhibitory effect on T cell proliferation by modulating the suppressive function of Treg cells. In addition, previous studied reported that RNF213 is the major susceptibility factor for Moyamoya disease[26,27], and plays a key role in resisting bacterial and virus infection[31,32]. All these indicate that the functions of RNF213 are complicated in the physiological regulation of the body.

FOXO1 belongs to a large family of forkhead box transcription factors. Nuclear localization of FOXO1/3a facilitates the expression of Foxp3 and enhances Treg cell stability[13]. Previous studies have demonstrated that FOXO1 serves as the substrate for ubiquitylation, facilitated by the E3 ubiquitin ligases SKP2, CHIP, and COP1[39–41]. Downregulation of Skp2 in diabetogenic autoreactive pathogenic T cells induces their conversion into Treg cells[42]. However, these E3 ubiquitin ligases primarily exert a negative regulatory effect on FOXO1 function by modulating K48-linked ubiquitination and subsequent degradation of FOXO1. In this study, we did not find that RNF213-regulated K48-linked ubiquitination and degradation of FOXO1. The K48-linked ubiquitination and protein level of FOXO1 showed no discernible difference between WT and Rnf213−/− Treg cells, whereas the K63-linked ubiquitination and nuclear localization of FOXO1 were significantly diminished in Rnf213−/− Treg cells compared to their WT counterparts (Supplementary Figs. 7A and 4G). The posttranslational modification of proteins, such as phosphorylation and acetylation, plays a crucial role in regulating the nuclear translocation of FOXO1[43]. Here, we have demonstrated that RNF213-mediated K63 ubiquitination also plays a crucial role in facilitating the nuclear translocation of FOXO1.

IFN-β represents the primary therapeutic approach for patients diagnosed with RRMS, effectively restoring the impaired suppression

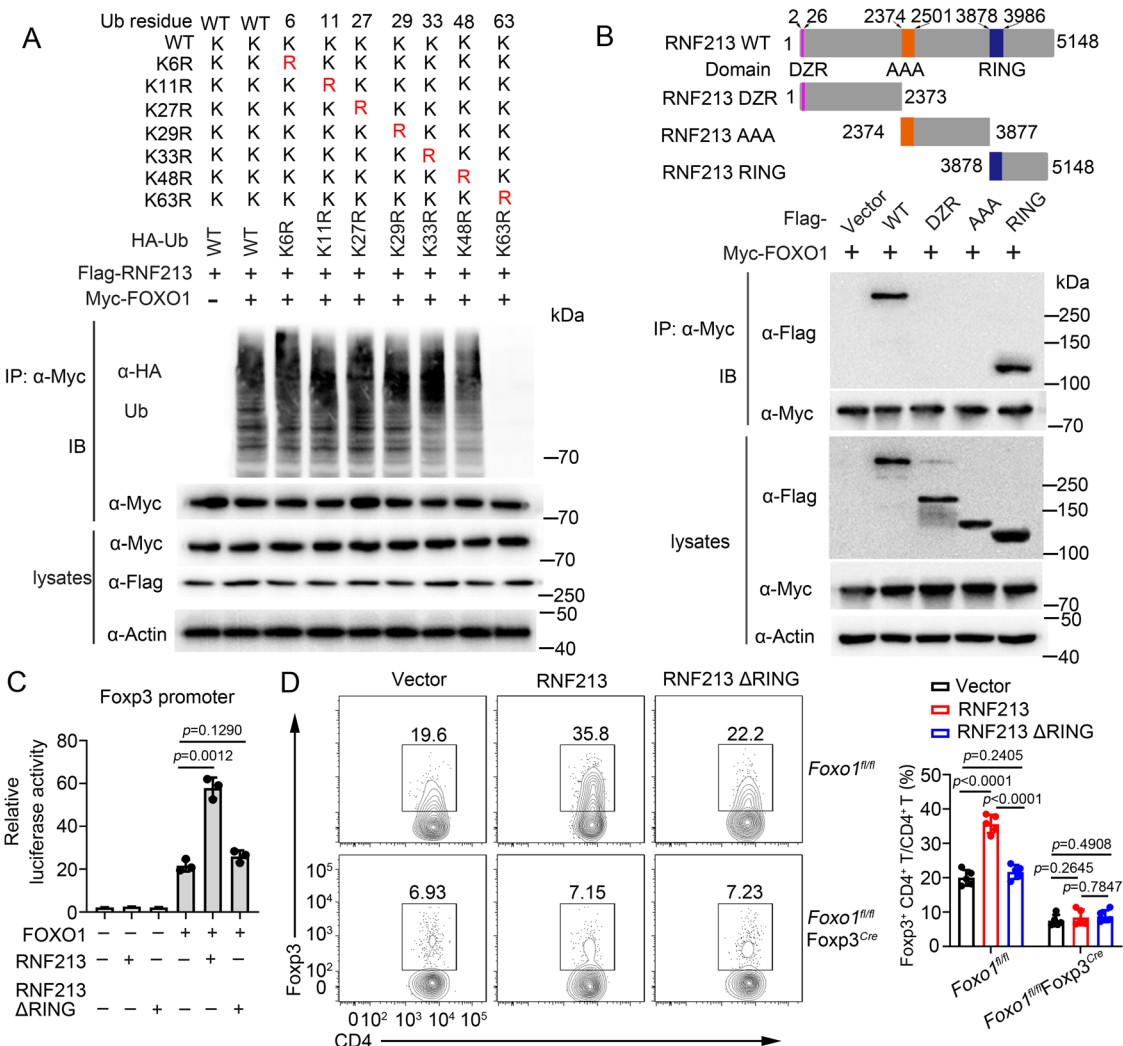

**Fig. 6 | RNF213 promotes FOXO1 K63-linked polyubiquitination and activity through Ubiquitination activity. A** HEK293T cells were co-transfected with a variety of ubiquitin mutants and other indicated plasmids for ubiquitination analysis. **B** Immunoprecipitation (IP) and immunoblot (IB) analysis of HEK293T cells that were transfected with indicated plasmids for 24 h. DZR double zinc ribbon, AAA ATPases associated with a variety of cellular activities, RING RING finger domain. **C** Luciferase activity of HEK293T cells transfected with a luciferase reporter driven by the Foxp3 promoter and expressing vector alone or FOXO1, RNF213, and RNF213ΔRING (lacking RING domain). **D** Flow cytometry of Foxp3 in CD4+ T cells infected with control retrovirus (Vector) or retrovirus expressing RNF213 or RNF213ΔRING differentiated under Treg-polarizing conditions. Pooled data are presented in the right panel. Data shown are the mean ± SD. *$P < 0.05$, **$P < 0.01$, and ***$P < 0.001$. $P$ values were calculated using a two-sided, unpaired Student's $t$-test in (**C, D**). N = 3 (**C**), n = 5 (**D**) repeats from three independent experiments. Source data are provided as a Source Data file.

of T cells associated with MS through enhanced induction of Treg cells both in vitro and in vivo[44–46]. In this study, we also observed that IFN-β treatment enhanced the differentiation of CD4+ T cells into Treg cells and mitigated the occurrence of EAE. The induction of RNF213 expression by IFN-β has been consistently demonstrated in multiple reports[30,32]. We demonstrated that IFN-β is able to induce RNF213 expression in CD4+ T cells. Moreover, we found that the effectiveness of IFN-β treatment for EAE was dependent on the induction of RNF213 expression and its role in facilitating Treg cell differentiation. RNF213 deficiency significantly decreased the ability of IFN-β to promote the development of Treg cells. Notably, we observed a significant correlation between the expression of RNF213 with both the onset of MS and response to IFN-β treatment. The expression of RNF213 in CD4+ T cells from MS patients who do not respond to IFN-β treatment was not induced by IFN-β. The aforementioned factor could potentially account for the lack of response to IFN-β observed in these MS patients. However, the mechanism of why RNF213 expression is not induced by IFN-β still needs to be further studied.

In summary, we identified RNF213 as a vital regulator of CD4+ T cell differentiation; it promoted Treg cell differentiation in a T cell-intrinsic manner through promoting K63-linked ubiquitination and nuclear translocation of FOXO1. Notably, RNF213 expression in CD4+ T cells was induced by IFN-β and exerts a crucial role in the therapeutic efficacy of IFN-β for MS. Together, our study findings collectively emphasize the pivotal role of RNF213 in modulating adaptive immune responses. RNF213 holds potential as a promising therapeutic target for addressing disorders associated with Treg cells.

## Methods
### Ethical approval
All animal experiments were conducted in accordance with protocols approved by the Medical Ethics Board and the Biosafety Management Committee of Southern Medical University. Informed consent was obtained in accordance with the Declaration of Helsinki and The Medical Ethics Committee of Zhujiang Hospital, Southern Medical

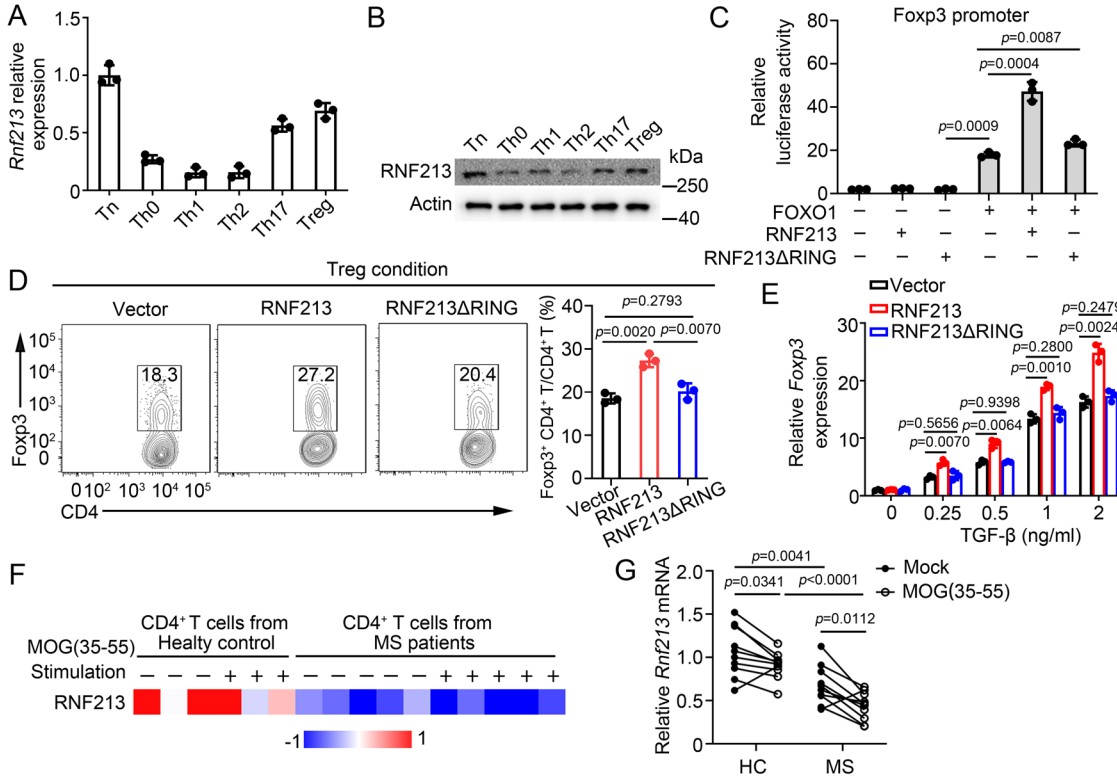

**Fig. 7 | RNF213 promoted Treg cell differentiation in human CD4⁺ T cells.**
**A**, **B** Purified human naïve CD4⁺ T cells were stimulated under standard Th0, Th1, Th2, Th17, or Treg conditions and harvested on day 5. RNF213 expression levels were detected by qPCR (**A**) or western blot (**B**). **C** Luciferase activity of HEK293T cells transfected with a luciferase reporter driven by the Foxp3 promoter and expressing vector alone or human FOXO1, RNF213 and RNF213ΔRING (lacking RING domain). **D** Flow cytometric analysis of intracellular human CD4⁺ T cells infected with a control retrovirus (Vector) or retrovirus expressing RNF213 or RNF213ΔRING and differentiated under standard Treg conditions. Pooled data are presented in the right panel. **E** QPCR analysis of Foxp3 in human CD4⁺ T cells

infected with a control retrovirus (Vector) or retrovirus expressing RNF213 or RNF213ΔRING and activated with the indicated amounts of TGF-β for 3 days. **F** The heatmap displaying RNF213 expression is based on RNA-seq data comparing CD4⁺ T cells stimulated with or without MOG(35-55) (GSE66763). **G** QPCR analysis of RNF213 in human CD4⁺ cells from healthy control (HC) (n = 10) and patients with multiple sclerosis (n = 10). Data shown are the mean ± SD. *$P < 0.05$, **$P < 0.01$, and ***$P < 0.001$. $P$ values were calculated using a two-sided, unpaired Student's $t$-test in (**C**–**E**), and was calculated using a paired Student's $t$-test in (**G**). N = 3 (**A**–**E**) repeats from three independent experiments. Source data are provided as a Source Data file.

University. Written informed consents were obtained from all participants for the use of PBMC samples.

## Mice
C57BL/6 mice (Wild type, WT) were from the Lab Animal Center of Southern Medicine University (NO. gdmlac9999, Guangzhou, China). RNF213-deficient (*Rnf213⁻/⁻*, Cat. NO. S-KO-12190), *Rnf213ᶠˡᵒˣ/ᶠˡᵒˣ* (*Rnf213ᶠˡ/ᶠˡ*, Cat. NO. S-CKO-13591), and *Foxo1ᶠˡ/ᶠˡ* (S-CKO-12115) mice were purchased from Cyagen Biosciences Inc. (Guangzhou, China). CD4-Cre (CD4ᶜʳᵉ, Cat. NO. SJ-022071) and *Foxp3ᶜʳᵉ* (Cat. NO. SJ-004337), CD45.1⁺ (Cat. NO. KI-210226) and *Rag1⁻/⁻* (Cat. NO. KI-00069) mice were purchased from the Shanghai Research Center for Model Organisms (Shanghai, China). *Rnf213ᶠˡ/ᶠˡ* mice were crossed with CD4ᶜʳᵉ or *Foxp3ᶜʳᵉ* mice to generate *Rnf213ᶠˡ/ᶠˡ*CD4ᶜʳᵉ or *Rnf213ᶠˡ/ᶠˡFoxp3ᶜʳᵉ* mice. *Foxo1ᶠˡ/ᶠˡ* mice were crossed with *Foxp3ᶜʳᵉ* mice to generate *Foxo1ᶠˡ/ᶠˡFoxp3ᶜʳᵉ* mice. *Rnf213ᶠˡ/ᶠˡFoxp3ᶜʳᵉ* mice were crossed with *Foxo1ᶠˡ/ᶠˡFoxp3ᶜʳᵉ* to generate *Rnf213ᶠˡ/ᶠˡFoxo1ᶠˡ/ᶠˡFoxp3ᶜʳᵉ*. RNF213-transgenic (*Rnf213ᵀᵍ*) mice were generated by Biocytogen (China). In brief, the pBS31- RNF213-Flag plasmid was co-electroporated with a recombinase expression vector into ES cells that were expressing the M2rtTA tetracycline-responsive transactivator under the control of the ROSA26 promoter. Transgene expression was induced by feeding the mice 2 mg/ml doxycycline in their drinking water supplemented with 10 mg/ml sucrose. To generate mice with Treg cell-specific over-expression of RNF213-Flag (*Rnf213ᵀᵍFoxp3ᶜʳᵉ* mice), *Rnf213ᵀᵍ* mice were crossed with Foxp3-Cre mice. All mice were all C57BL/6 background and

maintained in the Lab Animal Center of Southern Medicine University under specific pathogen-free (SPF) conditions. Animals were in a 12-h light/dark cycle beginning at 7 am (light) and 7 pm (dark). The ambient temperature was 20–22 °C with humidity 40–60%. Mice were euthanized using $CO_2$ asphyxiation dispensed from a fixed pressure regulator and inline restrictor controlling gas flow. $CO_2$ flow was maintained for at least 5 min. Death was verified following euthanasia by monitoring cessation of heartbeat and respiration, as well as a toe pinch reflex. All mice were used at an age of 6–12 weeks and were randomly divided into different groups. Experimental and control mouse groups were co-housed. Female mice were used for all experiments unless otherwise stated.

## EAE model
WT, *Rnf213⁻/⁻*, *Rnf213ᶠˡ/ᶠˡ*CD4ᶜʳᵉ, *Rnf213ᶠˡ/ᶠˡFoxp3ᶜʳᵉ*, and recipient *Rag1⁻/⁻* mice reconstituted by CD4⁺ T cells were immunized subcutaneously with 200 μg MOG(35-55) peptide emulsified in CFA (Difco Laboratories, USA) with 400 μg *Mycobacterium tuberculosis* H37Ra on day 0. To induce EAE development and assess the severity of EAE, mice also received 200 ng of pertussis toxin (Sigma, USA) by intraperitoneal injection on days 0 and 2. Symptoms of EAE were monitored daily using a classical clinical score ranging from 0 to 5 as follows: 0, no disease; 1, tail paralysis; 2, weakness of hind limbs; 3, paralysis of hind limbs; 4, paralysis of hind limbs and severe hunched posture; 5, moribund or death, as previously described[47].

## MOG(35-55) recall assay

Splenocytes or cells from the Central nervous system (Spinal cord and brain) were isolated from mice-induced EAE, were re-stimulated with 100 μg/ml MOG(35-55) in complete RPMI1640 media for 6 h to perform flow cytometry analysis of intracellular IFN-γ, IL-4, or IL-17A; re-stimulated for 48 h to perform Enzyme-linked immunosorbent assay (ELISA), as previously described[25].

## Mouse Naïve T cell isolation and T cell activation assay in vitro

Spleen and lymph node cells were isolated from mice. CD4+ T cells were negatively selected using EasySepTM. Mouse Naive CD4+ T cell Isolation Kit (Miltenyi, Germany). Purified naive T cells were stimulated with plate-bound anti-CD3 (1 μg/ml or indicated concentrations) and soluble anti-CD28 antibodies (1 μg/ml) in replicate wells of 96-well plates (1 × 10⁵ cells per well) for flow cytometry analysis and ELISA, 12-well plates (1 × 10⁶ cells per well) for qPCR and 6-well plates (5 × 10⁶ per well) for Western blot assays.

## Human T cell isolation and T cell activation assay in vitro

Human Peripheral blood mononuclear cells were isolated from the peripheral blood of healthy donors by Ficoll centrifugation. Human naïve CD4+ T cells were negatively selected using EasySepTM. Human naive CD4+ T cell Isolation Kit (Miltenyi, Germany). Purified human naive T cells were stimulated with plate-bound anti-CD3 (1 μg/ml or indicated concentrations) and soluble anti-CD28 antibodies (1 μg/ml) in replicate wells of 96-well plates (1 × 10⁵ cells per well) for flow cytometry analysis and ELISA, 12-well plates (1 × 10⁶ cells per well) for qPCR and 6-well plates (5 × 10⁶ per well) for western blot assays. Human CD4+ T cells from healthy patients with multiple sclerosis were selected using EasySepTM. Human CD4+ T cell Isolation Kit (Miltenyi, Germany). These CD4+ T cells were used for qPCR. We identified MS patients as IFN-β-responsive or non-responsive according to the previous description[48].

## Flow cytometry analysis

For intracellular cytokine staining assays, T cells isolated from the spleen or nervous system of mice, or from in vitro cultures were stimulated for 1.5 h with 100 mg/ml MOG(35-55) or PMA (50 ng/ml, Thermo Fisher Scientific, USA) and ionomycin (500 ng/ml, Thermo Fisher Scientific), before Brefeldin A (10 μg/ml, eBioscience, USA) was added to the culture for 3.5 h more. As previously described[49], for surface staining, cells were harvested, washed, and stained for 30 min on ice with mixtures of fluorescently conjugated mAbs or isotype-matched controls. For intracellular cytokine staining (ICS), cells were stained for surface molecules, fixed 20 min in IC Fixation buffer (Thermo Fisher Scientific), and incubated for 1 h in permeabilization buffer (Thermo Fisher Scientific) with appropriate mAbs of mice. Antibodies used in this study are listed in Supplementary Table 1. Cell phenotype was analyzed by flow cytometry on a flow cytometer (BD LSR II) (BD Biosciences, USA) or Attune NxT (Thermo Fisher Scientific). Data were acquired as the fraction of labeled cells within a live-cell gate and analyzed using FlowJo software (Tree Star). All gates were set based on isotype-matched control antibodies.

## CFSE T cell proliferation assay

Purified naïve T cells were labeled with 2.5 μM CFSE and then 5 × 10⁴ T cells/well were stimulated with anti-CD3/CD28. T cells were cultured for 72 h and proliferation was determined by flow cytometry analysis of CFSE dilution.

## CD4+ T cell differentiation

Purified human or mouse naive CD4+ T cells were stimulated with plate-bound anti-CD3 (2 μg/ml) and anti-CD28 (2 μg/ml) alone (Th0) or under Th1 (10 ng/ml IL-12 and 10 μg/ml anti-IL-4, Peprotech, USA), Th2 (20 ng/ml IL-4 and 10 μg/ml anti-IFN-γ, Peprotech), Th17 (2.5 ng/ml

TGF-β, 15 ng/ml IL-6, 10 μg/ml anti-IFN-γ and 10 μg/ml anti-IL-4, Peprotech) and Treg (1.5 ng/ml TGF-β, 10 μg/ml anti-IFN-γ and 10 μg/ml anti-IL-4, Peprotech) conditions. After 5 d of stimulation, the cells were subjected to Flow cytometry analysis, ELISA, or qPCR analyses.

## Enzyme-linked immunosorbent assay (ELISA)

Cytokine production in supernatants of in vitro cell cultures or sera of mice was measured by ELISA to assess the level of IFN-γ, IL-17, GM-CSF, and IL-10 (ExCell Bio, China) according to the manufacturer's protocol.

## Quantitative PCR (qPCR) analysis

Total RNA was isolated with Trizol (Thermo Fisher Scientific) according to the manufacturer's instructions. 100 ng⁻¹ mg of RNA was reverse transcribed to cDNA with random RNA-specific primers using the high-capacity cDNA reverse transcription kit (Applied Biosystems, USA). An Eppendorf Master Cycle Realplex2 and a SYBR Green PCR Master Mix (Applied Biosystems) were used for real-time PCR (40 cycles). The primer sequences used for PCR are in Supplementary Table 2.

## Competitive T cell transfers

Spleens or lymph nodes were removed from naïve WT (CD45.1+) and *Rnf213⁻/⁻* (CD45.2+) mice, and CD4+ T cells were negatively selected using EasySepTM. 1 × 10⁶ WT CD45.1+ CD4+ T cells and 1 × 10⁶ *Rnf213⁻/⁻* CD45.2+ CD4+ T cells were mixed together and co-transferred into *Rag1⁻/⁻* female mice via tail vein injection. One day later, the recipient mice were subjected to EAE induction.

## CD4+ T and Treg cell adoptive transfer in EAE

Spleens or lymph nodes were removed from naïve WT and *Rnf213⁻/⁻* mice and CD4+ T cells were negatively selected using EasySepTM. CD4+ T cells (1.0 × 10⁶ per mouse) were injected via tail vein injection into *Rag1⁻/⁻* female mice. One day later, the recipient mice were subjected to EAE induction. For evaluating the function of Treg cells in suppressing EAE, WT, and *Rnf213⁻/⁻* mice were immunized subcutaneously with 200 μg MOG(35-55) peptide emulsified in CFA with 400 μg *Mycobacterium tuberculosis* H37Ra on day 0. Spleens or lymph nodes were harvested on day 12. Treg cells (CD4+ CD25+) were sorted by FACS. Subsequently, CD45.1+ or 2D2 CD4+ T cells alone (5 × 10⁵ per mouse), CD45.1+ or 2D2 CD4+ T cells (5 × 10⁵ per mouse) with WT or *Rnf213⁻/⁻* Treg cells (2.5 × 10⁵ per mouse) were transferred into *Rag1⁻/⁻* mice via tail vein injection into. One day later, the recipient mice were induced EAE as described above.

## Plasmid constructs and transfection

Recombinant vectors encoding Flag-RNF213 (or RNF213ΔRING, or other indicated mutants), were cloned into the pGL4.14 (Sangon Biotech, China), and Myc-FOXO1 (or indicated mutants) and HA-ub were cloned into the pcDNA3.1(Sangon Biotech, China). The same or indicated quality plasmids were transfected into HEK293T cells with Lipofectamine 2000 (Invitrogen) according to the manufacturer's instructions

## Mass spectrometry

Proteins from Treg cells isolated from WT or *Rnf213^{Tg}Foxp3^{Cre}* mice were coimmunoprecipitated with anti-Flag antibodies. The immunoprecipitated proteins were eluted with buffer containing 8 M urea, 50 mM Tris (pH 8.0), reduced with 5 mM DTT, and alkylated with 15 mM iodoacetamide, followed by Liquid chromatography–mass spectrometry (LC-MS). Peptides were identified with a target-decoy approach using a combined database consisting of reverse protein sequences of the database. Up to two missed cleavages were allowed. Peptide identifications were reported by filtering reverse and contaminant entries and assigning to leading razor protein. Peptide inference and protein identification were filtered to a maximum of 1% and 1% false discovery rate, respectively. Data processing and statistical analysis were performed on Perseus (Version 1.6.0.7). A two-

sample *t*-test statistics was used with a *P* < 0.05 to report statistically significant expression.

## Nuclear and cytoplasmic extraction

Nuclear and cytoplasmic extraction were performed by NE-PER Nuclear and Cytoplasmic Extraction Reagents (Thermo Fisher Scientific) according to the manufacturer's protocol.

## Immunoblot, co-immunoprecipitation and ubiquitination assays

The experiments were performed as previously described[50]. Spinal cords or cells were washed three times with ice-cold PBS and then lysed in Nonidet P-40 lysis buffer containing 150 mM NaCl, 1 mM EDTA, 1% Nonidet P-40, and 1% protease and phosphatase inhibitor cocktail (Biotool). Equal amounts (20 mg) of cell lysates were resolved using 5 ± 15% polyacrylamide gels transferred to the PVDF membrane. Membranes were blocked in 5% non-fat dry milk in PBST and incubated overnight with the respective primary antibodies at 4 °C. The membranes were incubated at room temperature for 1 h with appropriate HRP-conjugated secondary antibodies and visualized with Plus-ECL (PerkinElmer, CA) according to the manufacturer's protocol. For immunoprecipitation assays, the lysates were immunoprecipitated with IgG or the appropriate antibodies and protein G Sepharose beads. The precipitates were washed three times with lysis buffer containing 500 mM NaCl, followed by immunoblot analysis. For deubiquitination assays, the cells were lysed with the lysis buffer, and the supernatants were denatured at 95 °C for 5 min in the presence of 1% SDS. The denatured lysates were diluted with lysis buffer to reduce the concentration of SDS below 0.1% followed by immunoprecipitation with the indicated antibodies. The immunoprecipitates were subjected to immunoblot analysis with anti-ubiquitin chains. Antibodies used in this study are listed in Supplementary Table 1.

## Immunofluorescence and microscopy

Sorted Treg cells were seeded in coverslips pretreated with polylysine, fixed with 4% PFA for 15 min at room temperature followed by 10 min of fixation with ice-cold methanol at −20 °C, washed twice with PBS and ice-cold methanol. Cells were permeabilized by using 0.1% saponin (Sigma-Aldrich) and stained with mouse FOXO1 or RNF213 antibody, followed by incubation with Alexa fluor® 555 anti-mouse IgG (1:500, Molecular Probes), Alexa fluor® 488 anti-rat IgG (1:250, Molecular Probes). DAPI staining occurred at a 1:1000 dilution for 5 min. Images of immunostained cells were taken using a Zeiss 810 confocal laser-scanning microscope.

## Retroviral packaging and transduction

Genes encoding wild-type RNF213, RNF213ΔRING, FOXO1, or FOXO1$^{K207R}$ were cloned into retroviral vector pMXs containing IRES-regulated GFP (Youbio, China), respectively. Each of the resulting plasmids was transfected into a packaging cell line, PLAT-T, using FuGENE6 (Roche, Switzerland). After incubation for 24 h, the culture supernatant was harvested and condensed as a viral stock. The CD4$^+$ T cells were stimulated by anti-CD3 and anti-CD28 antibodies for 24 h. The cells were then infected with retrovirus in the presence of 0.5 μg/ml of polybrene for 24 h and cultured further in the presence of 30 U/ml of IL-2 for 3 days. The cells were washed with fresh media and were stimulated with plate-bound anti-CD3 and anti-CD28 under Treg conditions. After 5 d of stimulation, the cells were subjected to Flow cytometry analysis, ELISA, or qPCR analyses.

## Statistics & reproducibility

All experiments were performed at least thrice. When shown, multiple samples represent biological (not technical) replicates of mice randomly sorted into each experimental group. The Investigators were not blinded to allocation during experiments and outcome assessment. Determination of statistical differences was performed with Prism 8 (Graphpad Software, Inc.) using unpaired two-tailed Student's *t*-tests (to compare two groups with similar variances).

## Reporting summary

Further information on research design is available in the Nature Portfolio Reporting Summary linked to this article.

## Data availability

The RNA-seq data supporting the findings of this study have been deposited in the Gene Expression Omnibus at the National Center for Biotechnology Information and will be available under accession numbers GSE195541 and GSE66763. Source data are provided with this paper.

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

## Acknowledgements

This work was funded by grants from the National Natural Science Foundation of China (32070906 awarded to S.H.F., 82201934 awarded to X.M.L.). Basic and Applied Basic Research Foundation of Guangdong Province (2023A1515010948 awarded to S.H.F.). Chinese Postdoctoral Science Foundation (2022M711505 awarded to X.F.Y.). Guangzhou Basic and Applied Basic Research Foundation (202201011028 awarded to X.M.L.). We thank all members of Shengfeng Hu' Lab for helpful dis-cussions and input.

## Author contributions

X.F.Y., X.T.Z., H.L.Z., X.M.L., and S.F.H. designed research; X.F.Y., X.T.Z., J.L.S., Y.L.F., D.N.N., X.L.Y., Y.T.C., X.D.Y., and Q.L. conducted research; X.F.Y., H.L.Z., X.M.L., and S.F.H. analyzed data; X.F.Y. and S.F.H. wrote the paper; H.L.Z., X.M.L., and S.F.H. provided essential reagents, or provided essential materials; H.L.Z., X.M.L., and S.F.H. as the corre-sponding author conducted the experiment. All authors read and approved the final manuscript.

## Competing interests

The authors declare no competing interests.
