## [Peer Review File · Nature Communications]

RNF213 promotes Treg cell differentiation by facilitating nuclear translocation of FOXO1 through K63-linked ubiquitinationREVIEWER COMMENTS

Reviewer #1 (Remarks to the Author):

In this manuscript the Authors address the role of the RNF213 ubiquitin ligase in the functional properties of CD4+ T cells in the context of the mouse model of Multiple Sclerosis (EAE) and in human primary CD4+ T cells. Their findings support that absence of RNF213 expression by CD4+ T cells worsen EAE development and attenuated Treg cell differentiation. Mechanistically, they show that RNF213 expression induces the nuclear translocation of FOXO1.

Although the mechanistic part of this study is well designed, it is not clear whether the Authors study Treg cell differentiation from CD4+ T cells (induced Tregs) or natural occurring Tregs (thymic). The first experiments appear to be focused on induced Tregs, however then they perform their Treg cell-related functional experiments with total Tregs from WT or KO animals which the majority are thymic Tregs.

Major comments

- What are the percentages of Foxp3+ Tregs that are induced upon transfer of WT and KO CD4+ T cells into RAG1-/- animals and immunized with MOG/CFA?
- In figure 2, upon co-transfer of CD4+ T cells, the authors do not observe differences in the levels of Th1 and Th17 (in contrast to Fig1). This indicated that the lack of differentiation of KO CD4+ T cells to Tregs does not affect the induction of Th1/Th17.
- It is critical that the Authors distinguish which Treg cells they study. The first two figures seem to refer to induced Tregs upon CD4+ T cell differentiation. In Figure 3 however, the study the Tregs from the RNF213KO mice which in my view is something different. Importantly, since KO mice at steady state do not show any pathologic signs or differences in Treg cell population.
- I am not sure whether through the results presented in Fig 6, the authors could make the following very strong statement "Together, these results suggested that the effectiveness of IFN- β treatment for EAE is dependent on the induction of RNF213 expression and its role in facilitating Treg cell differentiation"

Minor comments

- Fig 1F mice received the KO CD4+ T cells developed more severe disease and not attenuated as mentioned by the Authors
- Figures in the text are mislabeled (lines 297-301)

Reviewer #2 (Remarks to the Author):

General Comments:

The authors comprehensively characterise the role of RNF213 in CD4+ T-cell and specifically in Treg differentiation. This study utilised multiple Rnf213 knockout mice, such as Rnf213 -/-, Rnf213 fl/fl CD4 cre and Rnf213 fl/fl Foxp3 cre mice. They induced EAE and demonstrated that both constitutive and conditional Rnf213 deficiency in CD4+ T-cells significantly promoted EAE onset. Using adoptive transfer and in vitro experiments, the authors confirmed that Rnf213 is essential for Tregs to suppress T-cell mediated EAE. They confirmed that Rnf213 promotes Foxp3 expression through K63-linked ubiquitination and nuclear translocation of FOXO1. Furthermore, IFN- β induced RNF213 expression in CD4+ T cells, suggesting that RNF213 could be a novel therapeutic target for MS. The authors conclude that IFN- β therapeutic benefits for MS are through the induction of Rnf213 expression in Treg which in turn facilitates Treg differentiation.

Strengths

Overall, the manuscript provides an extensive analysis of the role of Rnf213 in CD4+ T cell and Treg differentiation. The authors performed a series of convincing experiments using various

conditional Rnf213 deficient mice and inducing EAE. They confirmed it is the loss of Rnf213 in Treg cells specifically, through both adoptive transfer experiments with CD4+ naive T cells from Rnf213-/- with or without Rnf213-/- Tregs and conditional knockout of Rnf213 in Foxp3+ cells, which promoted EAE. Furthermore, they provide comprehensive evidence to show that Rnf213 promotes Foxp3 expression through K63-linked ubiquitination and nuclear translocation of FOXO1.

Weaknesses

The current manuscript is confusing due to the presence of excessive data which is not directly related to the claims of the manuscript. The authors should consider removing certain data from the paper or moving it into supplementary.

The authors claim that both constitutive and conditional Rnf213 deficiency has no effect on thymic T cell development and T cell peripheral differentiation. They should provide more evidence to support this claim.

Major points:

Overall the authors support their claims. However, the figures and manuscript could be reorganised due to presence of excessive data to improve clarity and readability. I suggest this as the new ordering of main Figures:

1. Figure 1 – moved into Supplementary
 2. Figure 2 – moved into Supplementary
 3. New Figure 1 – Supplementary Figure 2
 4. Figure 3
 5. Figure 4
 6. Figure 5
 7. Figure 6 & 7 – either combine these figures into 1 Figure or move this data out of the paper
- The last two figures of the paper do not directly relate to the author's claims that Rnf213 promotes Foxp3 expression through K63-linked ubiquitination and nuclear translocation of FOXO1. I suggest that these figures either be combined into one figure/moved into Supplementary or moved out of the paper.

Additionally in the discussion, the authors do claim that there is no effect of Rnf213 deficiency in CD4+ T cell differentiation. Can they either reference previous studies or show equivalent experiments with the constitutive deficiency as they did with the conditional in Supplementary Figure 1? For both constitutive and conditional deletion, could they please show percentages/numbers of naive T cells in the spleen.

In general, authors should be also more careful with their claim that Rnf213 deficiency has no effect on thymic development as they only did a superficial characterisation of DN, DP, and SP. Could they either change the wording of this claim or show further evidence supporting this claim. Group sizes appear very small in certain figures and it is claimed that data are representative of three independent experiments however there are only n=3 in each group. It appears as if it is only one experiment? Does each dot represent an individual mouse? Please clarify this. Otherwise, consider increasing the group size as that is a low number of replicates.

Minor points:

Can authors please define clearly which markers they are using for each cell type in first section of results. Can they also please define clearer which mouse models they are using. There are some inconsistencies with how they refer to each mouse model. For example, please define what Rnf213 transgenic is.

Line 121 – instead of attenuated, it should say promoted EAE.

Line 125 – typo, lowercase Th17

Line 141 – significant decrease instead of increase.

Line 166 – remove “the”

Line 174-175 – reword this sentence for clarity.

Line 189 – significantly increased not diminished.

Line 194 – can authors please clarify why they used 2D2 mice in this particular experiment only.

Supplementary Fig 5C – clarify terminology in figure of what is the Rnf213 transgenic mouse and also use the same terminology in the text (Line 234-235)

Line 267 - clarify what Rnf213 tg is.

Line 300-301 – explain this experiment clearer in the text.

Line 355 – refer to Supplementary Figure 1 instead of 9.
Line 354-357 - please reword this section to make your claims clearer.
Line 366 – typo: RNF213 instead of NNF213
Line 380 – clarify which figures are being referred to
Line 390 – reword this sentence to make this clearer
Line 396 – response instead of respond
Supplementary Figure 6 – typo RNF157 instead of RNF213.

RESPONSE LETTER

Apr 21, 2024

Dear Editors and reviewers:

The authors wish to thank the reviewers and editors for the helpful suggestions for revision of the above manuscript. We have revised the manuscript (revised areas are highlighted in the version with highlights changes) per reviewers. Enclosed below is a point-by-point response to each.

Responses to Reviews for Manuscript Number: NCOMMS-23-61913A

Reviewer #1:

We thank Reviewer #1 for the time, effort and recognition given to the manuscript. According to your nice suggestions, we have made corrections to our previous draft, the detailed corrections are listed below.

Question: Major 1. What are the percentages of Foxp3⁺ Tregs that are induced upon transfer of WT and KO CD4⁺ T cells into RAG1^{-/-} animals and immunized with MOG/CFA?

Answer: We thank Reviewer #1 for the comment. We detected the percentages of Foxp3⁺ Tregs upon transfer of WT and *Rnf213*^{-/-} CD4⁺ T cells into *Rag1*^{-/-} mice in Supplementary Fig. 2I, and the results showed that *Rag1*^{-/-} mice receiving *Rnf213*^{-/-} CD4⁺ T cells had a lower proportion of Treg cells than those that received WT CD4⁺ T cells. They're about 16 percent and 7 percent, respectively (Supplementary Fig. 2I).

Question: Major 2. In figure 2, upon co-transfer of CD4⁺ T cells, the authors do not observe differences in the levels of Th1 and Th17 (in contrast to Fig1). This indicated that the lack of differentiation of KOCD4 T cells to Tregs does not affect the induction of Th1/Th17.

Answer: We thank Reviewer #1 for the comment. In Fig 1, we compared the differences of Th1 and Th17 in the WT and *Rnf213*^{-/-} mice (Supplementary Fig. 2 in revised version). Because the difference in the differentiation ratio of Treg will affect the differentiation of Th1 and Th17, the differentiation of Th1 and Th17 will be different. Thus, we carried out the competitive adoptive CD4⁺ T cell transfer assays to explore the intrinsic role of RNF213 in the CD4⁺ T cell differentiation. In the same environment, RNF213 deficiency in CD4⁺ T cells no longer affected Th1 and Th17 differentiation, but still affected Treg differentiation. These results demonstrated that RNF213 plays a specific role in promoting the differentiation of Tregs within CD4⁺ T cell population.

Question: Major 3. It is critical that the Authors distinguish which Treg cells they study. The first two figures seem to refer to induced Tregs upon CD4⁺ T cell differentiation. In Figure 3 however, the study the Tregs from the RNF213KO mice which in my view is something different. Importantly, since KO mice at steady state do not show any pathologic signs or differences in Treg cell population.

Answer: We thank Reviewer #1 for the comment. **(a).** In this manuscript, we mainly studied periphery Treg (pTreg, *in vivo*) and induced Treg (iTreg, *in vitro*), which are all differentiated from naive cells induced by TGF- β and IL-2. We have added the introduction of these Treg cells (Lines 61-66 revised areas are highlighted). **(b)** RNF213 deficiency does make a significant difference in the differentiation of pTreg in normal homeostasis and pathological conditions (EAE) in the spleen. RNF213 deficiency does not affect the proportion of pTreg in the spleen in homeostasis (Supplementary Fig. 1I and 3I), but decreased the pTreg differentiated in pathological conditions (EAE) in the spleen (Fig. 1D and Supplementary Fig. 2D). We are also perplexed by this matter. Our speculation is that it may be attributed to the substantial disparity in immune regulation intensity between homeostasis and pathology. Some research has demonstrated a comparable phenomenon (Chou WC *et al.* AIM2 in regulatory T cells restrains autoimmune diseases. *Nature*. 2021 591(7849):300-305). **(c)** The experimental steps depicted in Fig 3 were too simple in our previous version. Because normal mice have small spleens and lymph nodes, it is difficult to get enough Treg cells. Thus, we immunized the mice with MOG(35-55) without the injection of PT to induce EAE, and then sorted Treg cells from spleens or lymph nodes by FACS. We refined our experimental details in the Materials and methods section (Lines 536-548 revised areas are highlighted).

Question: Major 4. I am not sure whether through the results presented in Fig 6, the authors could make the following very strong statement “Together, these results suggested that the effectiveness of IFN- β treatment for EAE is dependent on the induction of RNF213 expression and its role in facilitating Treg cell differentiation”

Answer: We thank Reviewer #1 for the comment. The therapeutic efficacy of IFN- β is significantly diminished following RNF213 deficiency, as demonstrated by our findings. We think this conclusion is correct. However, in order to avoid any doubt, we have rearranged the results and conclusions of this section. We have changed it to another statement ‘RNF213 facilitates the differentiation of Treg cells and exerts a crucial role in the therapeutic efficacy of IFN- β for MS’ (Lines 348-350 revised areas are highlighted).

Question: Minor 1. Fig 1F mice received the KO CD4⁺ T cells developed more severe disease and not attenuated as mentioned by the Authors.

Answer: We thank Reviewer #1 for the comment. This is our mistake. We have revised it (Lines 132 revised areas are highlighted).

Question: Minor 2. Figures in the text are mislabeled (lines 297-301).

Answer: We thank Reviewer #1 for the comment. This is our mistake. We have revised them (Lines 327-328).

Reviewer #2:

We feel great thanks for Reviewer #2's professional review work on our article. As you are concerned, there are several problems that need to be addressed. According to your nice suggestions, we have made extensive corrections to our previous draft, the detailed corrections are listed below.

Question Major 1. Overall the authors support their claims. However, the figures and manuscript could be reorganised due to presence of excessive data to improve clarity and readability. I suggest this as the new ordering of main Figures:

1. Figure 1 – moved into Supplementary;
2. Figure 2 – moved into Supplementary
3. New Figure 1 – Supplementary Figure 2
4. Figure 3
5. Figure 4
6. Figure 5
7. Figure 6 & 7 – either combine these figures into 1 Figure or move this data out of the paper

The last two figures of the paper do not directly relate to the author's claims that Rnf213 promotes Foxp3 expression through K63-linked ubiquitination and nuclear translocation of FOXO1. I suggest that these figures either be combined into one figure/moved into Supplementary or moved out of the paper.

Answer: We thank Reviewer #2 for the suggestion. Based on the suggestion and study's logic, we have reorganized the figures and manuscript to enhance clarity and readability (including Figures 1-7 and Supplementary Figures 1-9).

1. Figure 1 and supplementary Figures 1-3: RNF213 played a vital role in CD4⁺ T cell differentiation and EAE pathogenesis *in vivo*.
2. Figure 2 and supplementary Figure 4: RNF213 specifically regulated Treg cell differentiation *in vivo* and *in vitro*.
3. Figure 3 and supplementary Figure 5: RNF213 deficiency in CD4⁺ T cells promoted autoimmunity by inhibiting immunosuppressive activity of Treg cells.
4. Figure 4 and supplementary Figure 6: RNF213 interacted with FOXO1 and facilitated the nuclear translocation of FOXO1.
5. Figure 5: RNF213-regulated Treg cell differentiation depends on FOXO1.
6. Figure 6 and supplementary Figure 7: RNF213 promoted Treg cell differentiation though regulating K63-linked ubiquitination of FOXO1.
7. Figure 7: RNF213 promoted Treg cell differentiation in human CD4⁺ T cells.
8. Supplementary Figure 8: The involvement of RNF213 is crucial in the therapeutic efficacy of IFN- β for EAE. (A combination of figure 6 and 7 from the previous version)
9. Supplementary Figure 9: Illustration of a model on RNF213-mediated regulation of Treg cell differentiation.

Question Major 2. Additionally in the discussion, the authors do claim that there is no effect of Rnf213 deficiency in CD4⁺ T cell differentiation. Can they either reference previous studies or show equivalent experiments with the constitutive

deficiency as they did with the conditional in Supplementary Figure 1? For both constitutive and conditional deletion, could they please show percentages/numbers of naive T cells in the spleen.

Answer: We thank Reviewer #2 for the suggestion. We conducted additional experiments and found that the deficiency of RNF213 did not have an impact on CD4⁺ T cell differentiation in homeostasis (Supplementary Fig. 1, Lines 112-116 revised areas are highlighted).

Question Major 3. In general, authors should be also more careful with their claim that Rnf213 deficiency has no effect on thymic development as they only did a superficial characterisation of DN, DP, and SP. Could they either change the wording of this claim or show further evidence supporting this claim. Group sizes appear very small in certain figures and it is claimed that data are representative of three independent experiments however there are only n=3 in each group. It appears as if it is only one experiment? Does each dot represent an individual mouse? Please clarify this. Otherwise, consider increasing the group size as that is a low number of replicates.

Answer: We thank Reviewer #2 for the comment. The initial statement did be somewhat imprecise, and we subsequently revised it with a different statement: RNF213-deficiency did not affect normal T cell homeostasis. Each dot represented an individual mouse. The consistent results obtained from three mice at a time led us to employ three mice per group in certain experiments. However, these experiments have been replicated at least three times, ensuring the accuracy of the findings. To avoid any misunderstanding, we have added a statement In ‘Statistics’ of ‘Materials and methods section’: All experiments were performed at least thrice. When shown, multiple samples represent biological (not technical) replicates of mice randomly sorted into each experimental group (Lines 621-623 revised areas are highlighted).

Question Minor 1. Can authors please define clearly which markers they are using for each cell type in first section of results. Can they also please define clearer which mouse models they are using. There are some inconsistencies with how they refer to each mouse model. For example, please define what Rnf213 transgenic is.

Answer: We thank Reviewer #2 for the comment. The previous version experienced some confusion in the nomenclature of mice, resulting in the emergence of different naming conventions for mice of the same species. In the revised version, we have implemented a standardized naming system for each type of mouse, ensuring consistency and providing comprehensive descriptions upon their initial introduction.

Question Minor 2. Line 121 – instead of attenuated, it should say promoted EAE.

Answer: We thank Reviewer #2 for the comment. This is our mistake. We have corrected it (Line 132 revised areas are highlighted).

Question Minor 3. Line 125 – typo, lowercase Th17.

Answer: We thank Reviewer #2 for the comment. This is our mistake. We have corrected it (Line 137 revised areas are highlighted).

Question Minor 4. Line 141 – significant decrease instead of increase.

Answer: We thank Reviewer #2 for the comment. This is our mistake. We have corrected it (Line 152 revised areas are highlighted).

Question Minor 5. Line 166 – remove “the”

Answer: We thank Reviewer #2 for the comment. This is our mistake. We have removed “the” (Line 177).

Question Minor 6. Line 174-175 – reword this sentence for clarity.

Answer: We thank Reviewer #2 for the comment. We have reworded this sentence (Line 186-187 revised areas are highlighted).

Question Minor 7. Line 189 – significantly increased not diminished.

Answer: We thank Reviewer #2 for the comment. This is our mistake. We have corrected it (Line 200 revised areas are highlighted).

Question Minor 8. Line 194 – can authors please clarify why they used 2D2 mice in this particular experiment only.

Answer: We thank Reviewer #2 for the comment. 2D2 mice belong to MOG(35-55)-specific T cell receptor transgenic mice, and adoptive transfer of its CD4⁺ T cells to *Rag1*^{-/-} recipients can spontaneously induce the generation of EAE phenotypes. We have added the statement in revised manuscript (Lines 205-207 revised areas are highlighted).

Question Minor 9. Supplementary Fig 5C – clarify terminology in figure of what is the Rnf213 transgenic mouse and also use the same terminology in the text (Line 234-235).

Answer: We thank Reviewer #2 for the comment. This is our mistake. The previous version experienced some confusion in the nomenclature of mice, resulting in the emergence of different naming conventions for mice of the same species. In the revised version, we have implemented a standardized naming system for each type of mouse (Fig 5A and Lines 249-259 revised areas are highlighted).

Question Minor 10. Line 267 - clarify what Rnf213 tg is.

Answer: We thank Reviewer #2 for the comment. We have added a statement for Rnf213 tg (Lines 229-230 revised areas are highlighted).

Question Minor 11. Line 300-301 – explain this experiment clearer in the text.

Answer: We thank Reviewer #2 for the comment. We have added a statement for Rnf213 tg (Lines 229-230 revised areas are highlighted).

Question Minor 12. Line 355 – refer to Supplementary Figure 1 instead of 9.

Answer: We thank Reviewer #2 for the comment. This is our mistake. The all-image references have been thoroughly revised.

Question Minor 13. Line 355 – refer to Supplementary Figure 1 instead of 9.

Answer: We thank Reviewer #2 for the comment. This is our mistake. We have revised it (Line 367-368 revised areas are highlighted).

Question Minor 14. Line 354-357 - please reword this section to make your claims clearer.

Answer: We thank Reviewer #2 for the comment. We have reworded this sentence (Lines 364-368 revised areas are highlighted).

Question Minor 15. Line 354-357 - please reword this section to make your claims clearer.

Answer: We thank Reviewer #2 for the comment. We have reworded this sentence (Lines 364-368 revised areas are highlighted).

Question Minor 16. Line 366 – typo: RNF213 instead of NNF213.

Answer: We thank Reviewer #2 for the comment. This is our mistake. We have revised it (Line 378 revised areas are highlighted).

Question Minor 17. Line 380 – clarify which figures are being referred to.

Answer: We thank Reviewer #2 for the comment. We have clarified the figure are being referred to (Lines 392-393 revised areas are highlighted).

Question Minor 18. Line 390 – reword this sentence to make this clearer.

Answer: We thank Reviewer #2 for the comment. We have reworded this sentence (Lines 401-402 revised areas are highlighted).

Question Minor 19. Line 396 – response instead of respond.

Answer: We thank Reviewer #2 for the comment. We have revised it (Line 408 revised areas are highlighted).

Question Minor 20. Supplementary Figure 6 – typo RNF157 instead of RNF213.

Answer: We thank Reviewer #2 for the comment. This is our mistake. We have revised it (Supplementary Figure 7 in revised revision).

We have answered each of the reviewers' comments in full, and anticipate that with incorporation of the suggested revisions, the manuscript will be judged appropriate for publication in *Nature communications*.

Sincerely yours,

Shengfeng Hu, Ph.D. & M.D.

Professor,
The Second Affiliated Hospital, The State Key Laboratory of Respiratory Disease,
Guangdong Provincial Key Laboratory of Allergy & Clinical Immunology,
Guangzhou Medical University,
#195, Dongfeng West Road, Guangzhou, Guangdong, P.R.China

REVIEWERS' COMMENTS

Reviewer #1 (Remarks to the Author):

The authors successfully addressed my comments

Reviewer #2 (Remarks to the Author):

Overall, the manuscript is now improved. However, there are still issues with text clarity, structure, and readability.

Important points to be improved:

1. The order of the text should be rearranged along with the figures.
2. The current title is confusing. Improve it to be more informative and reflective of the manuscript's primary findings.
3. Improve clarity of texts throughout the manuscript, including abstract.
4. Carefully proofread for grammatical errors.

REVIEWER COMMENTS

Reviewer #1 (Remarks to the Author):

The authors successfully addressed my comments

Reviewer #2 (Remarks to the Author):

Overall, the manuscript is now improved. However, there are still issues with text clarity, structure, and readability.

Important points to be improved:

1. The order of the text should be rearranged along with the figures.
2. The current title is confusing. Improve it to be more informative and reflective of the manuscript's primary findings.
3. Improve clarity of texts throughout the manuscript, including abstract.
4. Carefully proofread for grammatical errors.

RESPONSE LETTER

Jun 05, 2024

Dear Editors and reviewers:

The authors wish to thank the reviewers and editors for the helpful suggestions for revision of the above manuscript. We have revised the manuscript (revised areas are highlighted in the version with highlights changes) per reviewers. Enclosed below is a point-by-point response to each.

Responses to Reviews for Manuscript Number: NCOMMS-23-61913A

Reviewer #1:

We thank Reviewer #1 for the time, effort and recognition given to the manuscript.

Reviewer #2:

We feel great thanks for Reviewer #2's professional review work on our article. As you are concerned, there are several problems that need to be addressed. According to your nice suggestions, we have made extensive corrections to our previous draft, the detailed corrections are listed below.

Question 1. The order of the text should be rearranged along with the figures.

Answer: We thank Reviewer #2 for the comment. Based on the suggestion and study's logic, we have rearranged the order of the text along with the figures. The order of the text and the figures is as follows:

1. Supplementary Figures 1-2: To analyze the function of RNF213 in CD4⁺ T cells by RNF213-deficiency mice and T cell adoptive transfer assays.
2. Supplementary Figures 3 and Figure 1: To further analyze the function of RNF213 in CD4⁺ T cells by mice with conditional RNF213 knockout in CD4⁺ T cells.

3. Figure 2 and supplementary Figure 4: RNF213 specifically regulated Treg cell differentiation *in vivo* and *in vitro*.
4. Figure 3 and supplementary Figure 5: RNF213 deficiency in CD4⁺ T cells promoted autoimmunity by inhibiting immunosuppressive activity of Treg cells.
5. Figure 4 and supplementary Figure 6: RNF213 interacted with FOXO1 and facilitated the nuclear translocation of FOXO1.
6. Figure 5: RNF213-regulated Treg cell differentiation depends on FOXO1.
7. Figure 6 and supplementary Figure 7: RNF213 promoted Treg cell differentiation though regulating K63-linked ubiquitination of FOXO1.
8. Figure 7: RNF213 promoted Treg cell differentiation in human CD4⁺ T cells.
9. Supplementary Figure 8: The involvement of RNF213 is crucial in the therapeutic efficacy of IFN- β for EAE. (A combination of figure 6 and 7 from the previous version)
10. Supplementary Figure 9: Illustration of a model on RNF213-mediated regulation of Treg cell differentiation.

Question 2. The current title is confusing. Improve it to be more informative and reflective of the manuscript's primary findings.

Answer: We thank Reviewer #2 for the suggestion. We have revised the title from 'IFN- β -induced RNF213 promoted Treg cell differentiation by promoting FOXO1 ubiquitination and nuclear' to 'RNF213 promotes Treg cell differentiation by facilitating nuclear translocation of FOXO1 through K63-linked ubiquitination' (Lines 1-3 revised areas are highlighted).

Question 3-4. Improve clarity of texts throughout the manuscript, including abstract. Carefully proofread for grammatical errors.

Answer: We thank Reviewer #2 for the suggestion and the manuscript was improved for the English language by a professional language editing service.

We have answered each of the reviewers' comments in full, and anticipate that with incorporation of the suggested revisions, the manuscript will be judged appropriate for publication in *Nature communications*.

Sincerely yours,

Shengfeng Hu, Ph.D. & M.D.

Professor,

The Second Affiliated Hospital, The State Key Laboratory of Respiratory Disease,
Guangdong Provincial Key Laboratory of Allergy & Clinical Immunology,
Guangzhou Medical University,

#195, Dongfeng West Road, Guangzhou, Guangdong, P.R.China